# Seawater Reverse Osmosis Performance Decline Caused by Short-Term Elevated Feed Water Temperature

**DOI:** 10.3390/membranes12080792

**Published:** 2022-08-18

**Authors:** Thomas Altmann, Paulus J. Buijs, Andreia S. F. Farinha, Vitor R. Proença Borges, Nadia M. Farhat, Johannes S. Vrouwenvelder, Ratul Das

**Affiliations:** 1Innovation and New Technology, ACWA Power, 41st Floor, The One Tower, Sheikh Zayed Road, Dubai P.O. Box 30582, United Arab Emirates; 2Water Desalination and Reuse Center (WDRC), Biological and Environmental Science and Engineering (BESE) Division, King Abdullah University of Science and Technology (KAUST), Thuwal 23955-6900, Saudi Arabia; 3KAUST ACWA Power Center of Excellence (KAPCOE), Thuwal 23955-6900, Saudi Arabia

**Keywords:** reverse osmosis, SWRO, specific energy consumption (SEC), membrane compaction, membrane permeability

## Abstract

The shortage of fresh water resources has made the desalination of seawater a widely adopted technology. Seawater reverse osmosis (SWRO) is the most commonly used method for desalination. The SWRO process is energy-intensive, and most of the energy in SWRO is spent on pressurizing the seawater to overcome the osmotic barrier for producing fresh water. The pressure needed depends on the salinity of the seawater, its temperature, and the membrane surface properties. Membrane compaction occurs in SWRO due to hydraulic pressure application for long-term operations and operating temperature fluctuations due to seasonal seawater changes. This study investigates the effects of short-term feed water temperature increase on the SWRO process in a full-scale pilot with pretreatment and a SWRO installation consisting of a pressure vessel which contains seven industrial-scale 8” diameter spiral wound membrane elements. A SWRO feed water temperature of 40 °C, even for a short period of 7 days, caused a permanent performance decline illustrated by a strong specific energy consumption increase of 7.5%. This study highlights the need for membrane manufacturer data that account for the water temperature effect on membrane performance over a broad temperature range. There is a need to develop new membranes that are more tolerant to temperature fluctuations.

## 1. Introduction

The importance of desalination with respect to humanity’s ability to produce high-quality fresh water sustainably and at a low cost cannot be overstated [1]. Two-thirds of the global population live under severe water scarcity for at least one month a year [2]. With its arid and semi-arid climate and limited freshwater resources, the Middle East relies mainly on desalination for sustenance [3,4,5,6,7,8,9]. Desalination is a proven technology that helps alleviate the water stress in these regions. Seawater reverse osmosis (SWRO) has gained the limelight as a promising desalination technology to source fresh water. SWRO holds roughly a 69% market share among all desalination technologies [10,11]. SWRO requires a high energy input to extract fresh water from seawater compared with the production from conventional sources such as rivers and lakes [12]. The specific energy consumption (SEC) of SWRO is the key parameter characterizing its performance [13,14]. SEC is the energy required in kWh to produce a m^3^ of product water. It comprises contributions from all sections of the SWRO plant, i.e., the seawater intake, pretreatment, RO desalination composed of high-pressure pumps, membranes, energy recovery devices, and the product post-treatment section. The RO section of the SWRO plant contributes between 60 and 80% of the total energy demand [12,13,14].

The energy consumed in the RO section depends primarily on the osmotic barrier of feedwater, which is determined by its salinity and the hydraulic water permeability of the membrane. The SEC generally benefits from membranes having higher permeability, selectivity, and tolerance to fouling [15]. Several studies focus on how SEC can be decreased by using membranes with a higher permeability [16,17,18,19,20,21]; a higher permeability allows for a lower pressure required to achieve the same permeate flux [22]. Membrane deformation or compaction due to increased hydraulic pressure and changes in feed water temperatures is observed in SWRO [23,24,25], which reduces the permeability of the membrane and requires the need for higher pressures to maintain a constant permeate flux [26]. Compaction is primarily irreversible [27], and membrane deformation has been verified through membrane autopsies after their use [25]. The fundamental behavior of compaction and its effect on SWRO performance remains poorly understood. Studies focusing on experimental compaction verification primarily address membrane deformation due to high hydraulic pressures [28,29,30,31]. Few studies report on the variations of SWRO performance due to expansion and shrinkage of membrane materials resulting from variations in feed water temperatures, as is the case for SWRO in the Middle East. The seawater temperatures in the Middle East vary seasonally due to the arid and semi-arid climate. Temperatures can reach as high as 38 °C in peak summers and as low as 19 °C in winters. Appendix A shows the typical annual variations of seawater temperatures for this region. Global warming and climate change have exposed the region to evident warming of seawater temperatures. It is estimated that seawater temperatures in these regions can increase anywhere between 0.02 and 0.07 °C/year [32,33,34]. The seasonal extremities in temperatures affect the performance of SWRO. Even pressurization of seawater to higher than 60 bar pressure increases its temperature due to the inefficiencies of the pumps [35]. It is known that high temperatures tend to impact membrane salt rejection and scaling negatively. In contrast, low temperatures require higher pressures (more energy) to achieve the same permeate water flux [36]. Due to the seasonal highs and lows in seawater temperatures in the region, the membranes are subjected to thermal stress, which causes their deformation, reducing permeability and hence affecting the SEC in the process [26]. Hence, a critical question remains to be answered, “*How do variations in seawater temperature affect the performance of SWRO membranes in the long term?*”.

In light of the aforementioned, in the present study, we analyze the performance of commercially available 8” Polyamide thin-film composite (TFC) spiral wound SWRO membranes operating in a full-scale SWRO pilot. The pilot test unit consisted of ultrafiltration and reverse osmosis fed by Red Sea seawater. To simulate the seasonal variations in seawater temperatures in the Middle East and their effect on the SEC of the SWRO process, we ran the pilot in recirculation mode and controlled the SWRO inlet seawater temperature.

## 2. Materials and Methods

### 2.1. Pilot Plant Details

The pilot plant was designed for seawater desalination research; it was equipped with state-of-the-art instrumentation and was fully automated through programmable logic controllers (PLC), which recorded all the signals into a data logger. All sensors and instruments used in the pilot plant are complied with industrial standards with a maximum error of ±0.2%. The details of different sections of the pilot are shown in Figure 1 and listed below:

**Seawater intake**—It was an open intake in the Red Sea, located 1.2 km from the coast and at a depth of 10 m. There was a 2 mm mesh at the mouth of the intake pipe to prevent the suction of larger objects and mussels. There was no chlorination at the intake.

**Microstrainer**—The seawater from the intake entered the pilot through a micro\strainer; it consisted of a 250 µm mesh that prevented large objects from interfering with the pilot’s operations.

**Seawater intake tank**—This was an intermediate tank to collect the seawater before ultrafiltration (UF).

**UF**—Dow Ultrafiltration™ SFP-2880 membranes were used in the UF modules. It was operated at a flux of 75 LMH for 60 min and then backwashed with the UF permeate at 100 LMH for 3 min. Chemically enhanced backwash was performed once daily. The filtrate from the UF was stored in the UF filtrate tank before being pumped to the SWRO system.

**RO section**—Water from the UF filtrate tanks went through a SWRO feed pump and then a high-pressure pump (HP pump) to the SWRO pressure vessels. It was a two-pass system; each pass had two 8- and one 16-inch pressure vessel. Permeate and brine from each pass could go to a tank for storage.

**Recirculation tank and cooler**—There was a provision for putting permeate and brine from the first pass into a recirculation tank. This could also be used as feed to the HP pump through the recirculation tank pump. The flow from the recirculation tank to the HP pump went through a cooler to control its temperature; it was cooled using a coolant, and a constant temperature was maintained through a PLC. Figure 1 shows a process flow diagram for the pilot.

### 2.2. Operating Conditions

To study the effect of seawater temperature on the performance of spiral wound SWRO membranes, we loaded one pressure vessel with 2 × 400 ft^2^ and 5 × 440 ft^2^ area 8” diameter elements. Chlorine-free seawater was fed into the pilot plant. Chlorination has been shown to cause higher microorganism growth [37]; we avoided chlorination for that purpose. No other chemicals were dosed after the test started. The pilot plant was operated in two phases. The first phase was for membrane performance stabilization. The plant was operated in a once-through mode during this phase at normal seawater conditions without any chemical dosing. The second phase was aimed at testing the membrane performance under varying temperatures. The plant was operated in recirculation mode during this phase. The seawater was heated through the HP pump, and its temperature was controlled using a water-cooled exchanger. The entire pilot test was conducted at 40% system recovery. The seawater conditions and the standard operating parameters are listed below.

### 2.3. Seawater Conditions

Temperature: unregulated, 24–35 °CpH—natural, 8.3Total dissolved solids (TDS)—natural, approx. 41,500 ppm

Table 1 below lists the elemental composition of the seawater used for the pilot determined analytically using an Agilent 7500cx Inductively coupled plasma mass spectrometry (ICP-MS) [38].

### 2.4. Operating Parameters

Feed temperature ~25–40 °C (after HP pump).Feed conductivity ~60.0 mS/cm.Feed flow ~7.5 m^3^/h.Recovery—40% constant.Permeate flow—3.0 m^3^/h constant and no permeate split operation.Brine flow—4.5 m^3^/h.Membrane flux calculated 2 × 400 + 5 × 440 ft^2^ and 3.0 m^3^/h = 10.75 LMH.Operation conditions were maintained by brine and feed control valves.SWRO inlet temperature during recirculation operation was maintained by circulating the seawater through a water-cooled exchanger, which was operated automatically through a PLC.

Seawater temperature, conductivity, oxidation reduction potential (ORP), and RO inlet temperature after the HP pump were continuously monitored during the test. Parameters required for membrane performance normalization were recorded in the distributed control system (DCS) at an interval of 30 s.

### 2.5. Experiment Conditions

This pilot test was primarily aimed at studying the effects of the seasonal extremities of seawater temperature on the performance of SWRO. The pilot was operated in a once-through mode for the first two weeks for membrane performance stabilization. Afterward, seawater recirculation was started to maintain a constant SWRO inlet temperature. The temperature was kept at 25 °C for two days for stabilization. Afterward, the SWRO inlet temperature gradually rose to 40 °C and operated at 40 °C for one week. The extreme temperatures in the summer season usually last for a week in the Gulf; therefore, the duration was selected to replicate these conditions. After a week of operation at 40 °C, the temperature gradually decreased to 25 °C, and the steady-state pilot operation at 25 °C continued under recirculation mode. A summary is shown in Table 2.

## 3. Results

This study looked at the effects of short-term feed water temperature increase on the SWRO process performance in a full-scale pilot containing pretreatment and an SWRO installation.

Figure 2 shows the pH and conductivity trends of the feed, permeate, and brine streams during the pilot run. Appendix A shows the oxidation reduction potential (ORP) of seawater. These measurements were done with handheld instruments to validate data from online sensors. The pH and conductivity trends seen in Figure 2 were as expected in a full-scale plant.

Figure 3 summarizes the entire pilot operation for 30 days. It shows the variation in normalized flux, feed pressure, and temperature during the pilot operation (a detailed normalization procedure is described in Appendix A). As mentioned earlier, the pilot was operated in two phases at a constant recovery of 40% (Appendix A). The first phase was to stabilize membrane performance and test the pilot equipment and instrumentation. The system stabilization was confirmed by the feed pressure and the permeate conductivity readings. During the second phase, the pilot was operated in a recirculation mode under controlled temperatures, which was maintained by the heat of the HP pump and a water-cooled heat exchanger. The permeate conductivity increased at a higher temperature (Figure 4); however, when the temperature was lowered to 25 °C, the conductivity was lower than during the conditioning phases, signaling membrane structural changes. Figure 5 below shows the operational data (feed pressure and salt rejection) during the recirculation operation; seawater feed temperature to the RO varied between 25 °C to 40 °C.

It can be seen from Figure 5A that a lower pressure was required to maintain a constant recovery at an elevated temperature. Permeate flux increased as the feed temperature increased. The viscosity of seawater decreased as temperature increased, and the water permeation rate through the membrane increased. Permeate flux typically increased with temperature linearly with viscosity [39,40,41]. When the RO feed temperature increased from 25 °C to 40 °C, the feed pressure reduced from 56.5 bar to 54.8 bar. On the contrary, when the RO feed temperature decreased from 40 °C to 25 °C, the feed pressure increased from 56.7 bar to 60.25 bar. The additional pressure buildup during the reduction in operating temperature resulted from operation at higher water temperature, possibly causing compaction of the membrane surface. Higher temperatures also increased the solubility of the solute, and a higher diffusion rate of the solute through the membrane was possible [41], causing a reduction in salt rejection by the membrane, as observed in Figure 5B. Figure 5B shows that when the RO feed temperature increased from 25 °C to 40 °C, salt rejection decreased. However, when the feed water temperature returned to 25 °C, we saw an increase in salt rejection by 0.2%. This improvement in salt rejection possibly results from membrane surface compaction, but it comes at the cost of additional energy expenditure, as discussed earlier. Salt rejection is calculated as follows:(1)SR(%)=(1−σpermeateσseawater)∗100
where *σ* denotes electrical conductivity.

Figure 6 below presents the normalized operation data for the pilot (a detailed normalization procedure is described in Appendix A).

The normalized data presented clearly shows a significant decrease in NPF, a reduction in NSP, and a constant increase in NDP. The continuous increase in normalized differential pressure can be attributed to the deposits on the membrane surface during the pilot run. Inductively coupled plasma mass spectrometry (ICP-MS, Agilent 7500cx) analysis was performed to check the major elemental composition of deposits on the membrane and feed spacer at the end of the 30-day study when the membrane element was autopsied. The deposits contained primarily Iron, Nickel, Magnesium, and Manganese (Table 3). A detailed elemental composition of the deposits on the feed spacer and membrane surface can be found in Appendix A.

From Figure 6, it can be seen that there was a 0.43 bar increase in normalized differential pressure (NDP), corresponding to a 2.5% decrease in normalized permeate flow (NPF) during the once-through operation mode. There was a subsequent increase of 0.3 bar in NDP and a corresponding 13.9% decrease in NPF during the recirculation operation mode, which raised the temperatures to 40 °C. Hence, the increase in NDP was certainly not the reason behind the steep decrease in NPF, which increased the pressure required to maintain the recovery rate. There was also a reduction of 0.2% in normalized salt passage (NSP) post recirculation operation, which points toward a possible membrane surface compaction. The changes in NPF and NSP are with reference to the initial conditions at the startup of the experiments; hence, they represent the actual change in the parameter rather than measurement errors.

The changes that occurred to the membrane surface due to high-temperature operation directly impacted the specific energy consumption (SEC) of the RO process (Figure 7). Additional energy (higher pressure) was required to maintain recovery ratios (constant permeate flux), as shown in Figure 7. When the RO inlet temperature was increased from 25 °C to 40 °C, a decrease of 2.5% in the SEC of the process was seen (Figure 7). The SEC decrease was due to an increase in the water permeation rate through the membrane resulting from a reduction in viscosity of the RO feed at higher inlet temperatures. However, when the feed temperature returned to 25 °C, we saw an increase in SEC of the process by 7.5%. It was expected that as the viscosity of the fluid returned to its original value of 25 °C, the SEC would increase back by the same amount of 2.5%, which was not the case. The additional energy required to maintain production was attributed to the possible compaction of the membrane surface, which resulted from surface deformations of the porous support due to thermal stress [42,43,44]. Figure 8 shows a scanning electron microscopy image of the used membrane after operation, where the thickness of the polyamide layer was observed to be 28 µm. This same virgin membrane had a polyamide layer thickness of 40 µm, as reported by [45], which was also confirmed by the membrane OEM. Compaction is primarily irreversible, and these structural changes permanently impact the energy consumption of the RO process [27].

## 4. Discussion

Structural changes in thin-film composite (TFC) polyamide RO membranes combined with changes in solvent viscosity and solute diffusivity govern the relationship between RO inlet temperature, mass transfer, membrane transport, and SEC of the RO process. Considering only changes in solute and solvent properties will misconstrue the impact of RO inlet temperature on the performance of RO membranes, which has been discussed theoretically by researchers [41,46]. The data from the pilot test suggested that the performance of TFC polyamide RO membranes is sensitive to changes, even temporarily, in the RO inlet temperature. Membrane manufacturers need to account for the structural changes in the membranes due to thermal stress and provide flux data for the modules over a wide temperature range up to 40 °C.

The results presented in this study underscore the importance of seasonal variations in the temperature of seawater and their potential impact on the SEC of a RO process (Figure 9). In a water-stressed region such as the Middle East, SWRO is widely adopted as a low-cost and sustainable clean, fresh water source for human use [3,4,8,9]. Therefore, a rigorous methodology considering all parameters that affect the SWRO process efficiency and its SEC is essential to optimize the final water tariff. The energy consumed by the RO section of an SWRO plant can account for ~40% of the total water cost of a desalination plant [12,47]. Hence, considering changes in the membrane structure is of utmost importance in accounting for its effect on the pressure required to achieve a specific permeate flux. Most desalination project contracts in the Middle East are offered under a BOO (Build, Own, Operate) structure, with a long concession period (~20–25 years) [8,48]. The contracts have technical guidelines to be met to achieve final water quality, and they also define the water tariff for the entire concession period. To avoid unforeseen changes in water cost during the concession period due to changes in energy consumption by the RO process, it is of paramount importance that all parameters that account for changes in feed pressure required to achieve a specific permeate flux are considered during project design, including the changes in membrane structure due to seasonal variations of seawater temperature. Concerted computational investigations are needed to develop models to account for increased energy requirements in RO due to membrane structural changes during their operation. However, such studies are beyond the scope of this work. The development of newer RO membranes with materials that can resist the effects of such thermal compaction at elevated temperatures up to 40 °C is also needed [42,47,48].

## 5. Conclusions

To summarize, this study provided an experimental verification for the impact of temperature variations on the energy consumption of an SWRO process. It provided insights into rationally designing new SWRO projects. The study was conducted at a full-scale pilot plant consisting of a seawater intake point, microstrainer, ultrafiltration, and an SWRO installation consisting of a pressure vessel containing seven industrial-scale 8′′ diameter spiral wound membrane elements.

Results showed that:A SWRO feed water temperature of 40 °C, even during a short period of 7 days, caused a permanent performance decline, as illustrated by a strong specific energy consumption (SEC) increase of 7.5%.A 7.5% increase in SEC, depending on the plant size, translates into an additional operating cost of USD 250,000 a year for a 60,000 m^3^/day production capacity plant to USD 2.5 M a year for a 600,000 m^3^/day capacity plant [47].There are financial consequences, in addition to contractual implications, for the use of additional energy. Since the energy required by the plant is defined during project development, it may require payment of an inflated tariff for the additional consumption.

The authors conclude with the hope that this study heralds coordinated efforts from the membrane manufacturers to provide insights into the changes in membrane structures due to temperature variation during RO operation, and that their subsequent effect on energy consumption is accounted for in membrane projection programs.

## Figures and Tables

**Figure 1 membranes-12-00792-f001:**
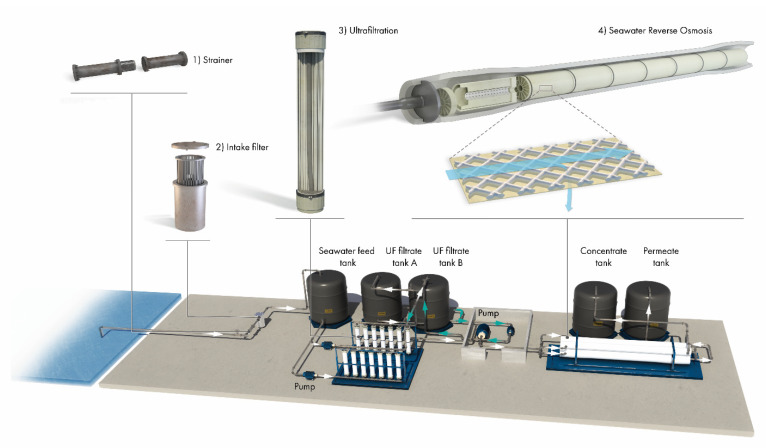
Schematic of the seawater reverse osmosis pilot used for the study, operated with seven 8” diameter spiral wound SWRO membrane modules in series.

**Figure 2 membranes-12-00792-f002:**
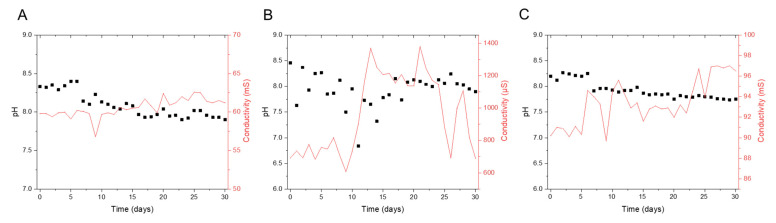
pH and conductivity profile of the (**A**) seawater feed, (**B**) SWRO permeate, and (**C**) SWRO brine during the 30-day pilot test study.

**Figure 3 membranes-12-00792-f003:**
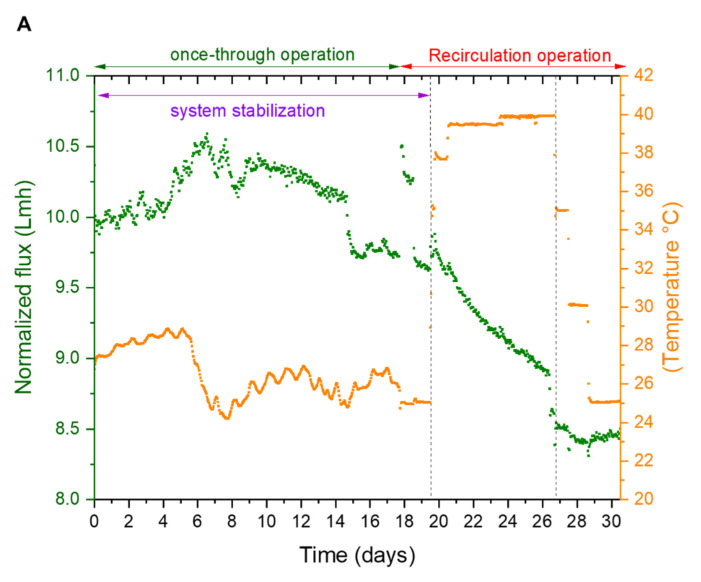
Operating data of the pilot test, data captured from the distributed control system (DCS). The plots (**A**) show the variation in normalized flux and (**B**) feed pressures with changes in feed temperature. The dotted vertical lines indicate the period where the feed water temperature was maintained at 40 °C.

**Figure 4 membranes-12-00792-f004:**
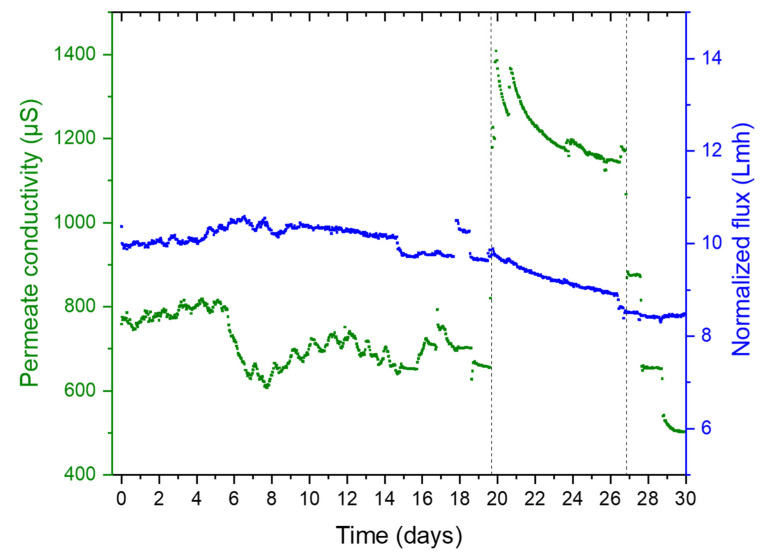
Variations in the permeate conductivity and flux during the pilot operation.

**Figure 5 membranes-12-00792-f005:**
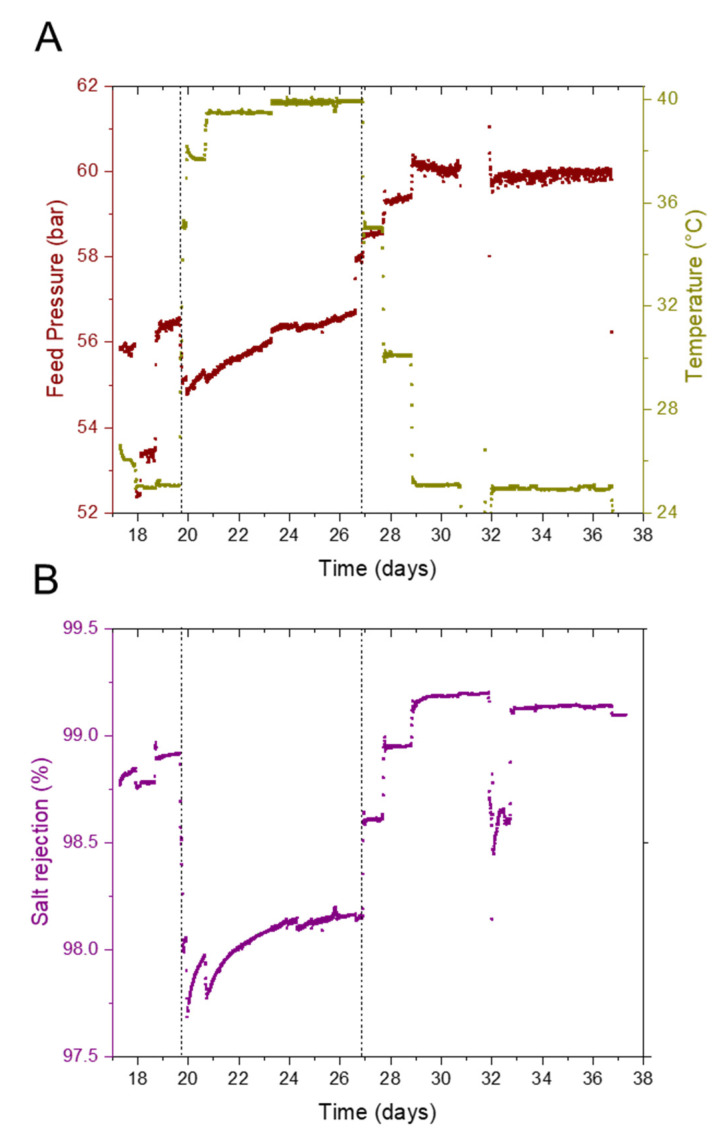
Pilot operating data during recirculation operation. (**A**) Variation in feed pressure with temperature (**B**) Variation in salt rejection with temperature under constant recovery of 40% and feed flow rate of 7.5 m^3^/h.

**Figure 6 membranes-12-00792-f006:**
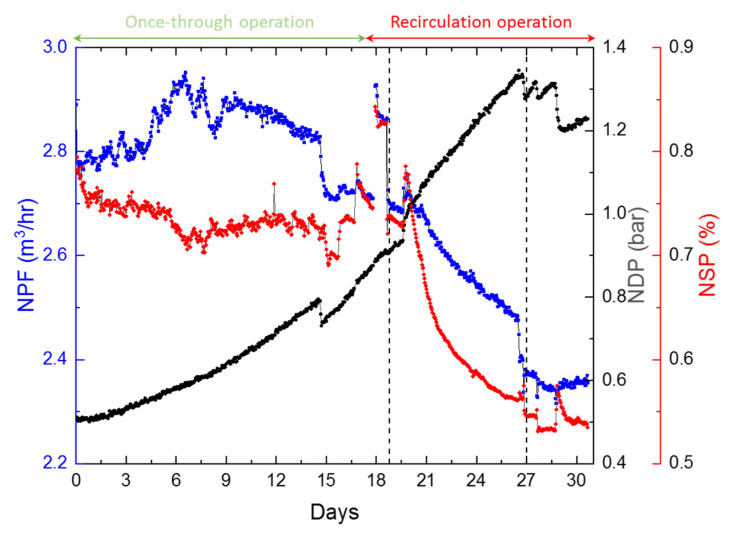
Normalized operation data for the pilot. NPF denotes the normalized permeate flow; NDP, the normalized differential pressure; and NSP, the normalized salt passage. Each line corresponds to the axis with similar color.

**Figure 7 membranes-12-00792-f007:**
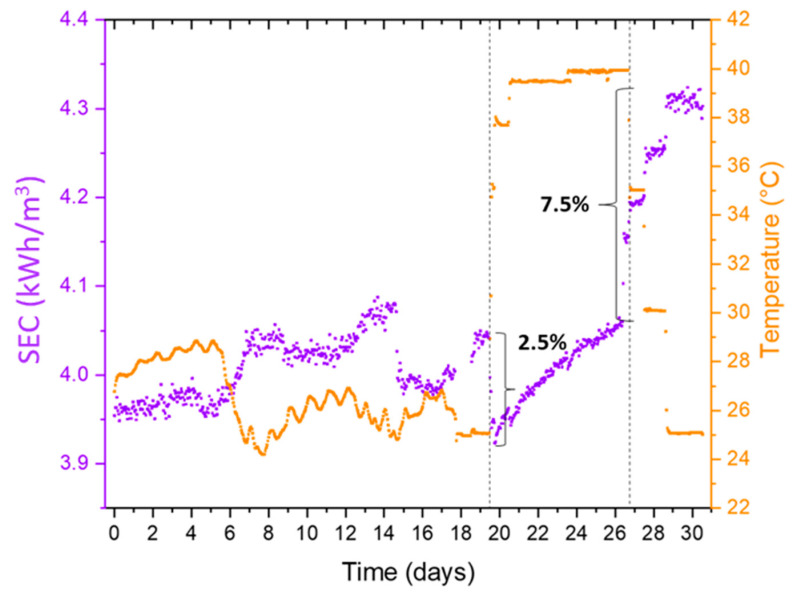
Variations in specific energy consumption (SEC) during the pilot test.

**Figure 8 membranes-12-00792-f008:**
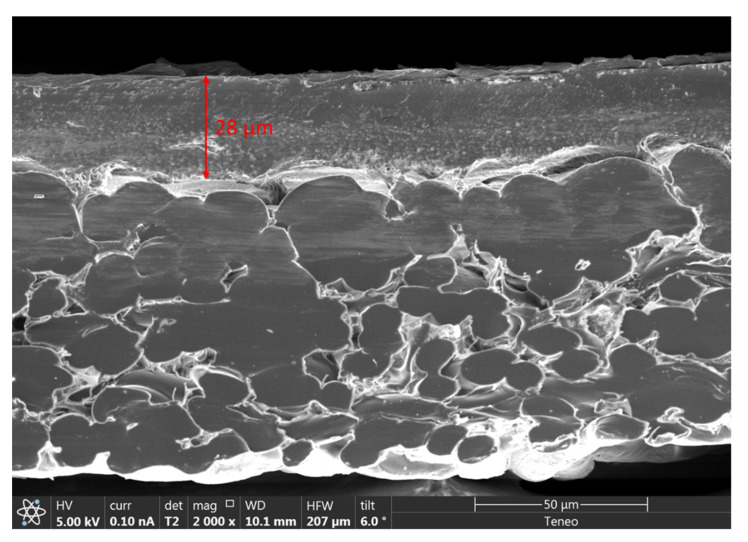
Scanning electron microscopy (SEM) image of the used membrane after operation.

**Figure 9 membranes-12-00792-f009:**
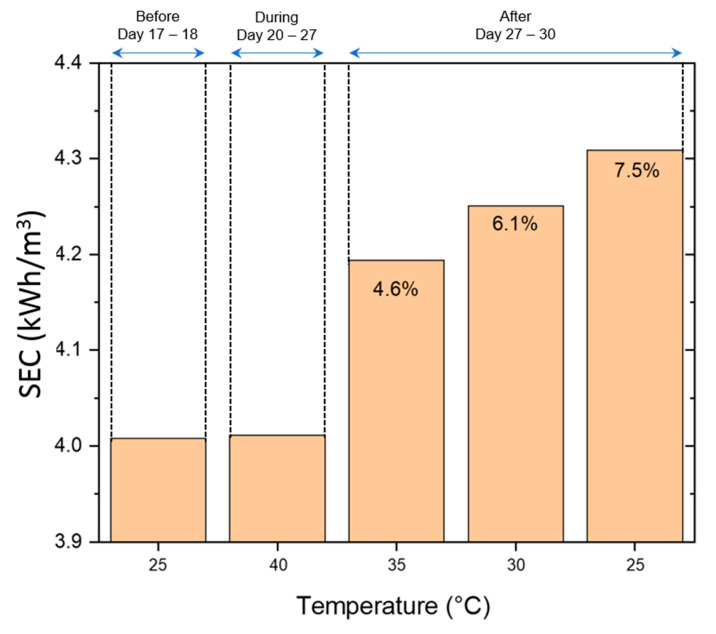
Average specific energy consumption (SEC) before, during, and after short-term (7-day) RO-feed seawater temperature increase. The change in SEC can also be correlated with the change in NPF, as discussed in Figure 6.

**Table 1 membranes-12-00792-t001:** Elemental composition of seawater used for the pilot test.

Cations		Unit	Value	Anions		Unit	Value
**Calcium**	Ca^++^	mg/L	485	**Bicarbonate**	HCO_3_^−^	mg/L	153
**Magnesium**	Mg^++^	mg/L	1649	**Chloride**	Cl^−^	mg/L	22,918
**Sodium**	Na^+^	mg/L	12,549	**Bromide**	Br^−^	mg/L	78.29
**Potassium**	K^+^	mg/L	464	**Sulfate**	SO_4_^−−^	mg/L	3220
**Strontium**	Sr^++^	mg/L	5.32	**Fluoride**	F^−^	mg/L	1.34
**Barium**	Ba^++^	mg/L	0.01	**Carbonate**	CO_3_^−−^	mg/L	6.0
				**Nitrate**	NO_3_^−^	mg/L	1.3
**Other parameters**							
**TDS**		mg/L	41,540	**Boron**	B	mg/L	4.9
**Conductivity**		µS/cm	59,800	**Silica**	Si	mg/L	1.4
**pH**		-	8.24	**TOC**	C	mg/L	3.0
**Temperature**	T	°C	25	**Density**	ρ	g/L	1028

**Table 2 membranes-12-00792-t002:** Pilot operating conditions during the different phases.

Phase	Operation Phase Description	Water Temperature (°C)	Time (day)	Operation Mode	
				**Once Through**	**Recirculation**
1a	Stabilization of membranes	Not regulated *	0 to 16	X	
1b		25	17–18		X
2	Temperature increase	From 25 to 40	19–20		X
3	Operation at high temperature	40	20–27		X
4	Temperature decrease reaching 25 °C	35, 30, and 25	27–30		X

* natural seawater temperature.

**Table 3 membranes-12-00792-t003:** The feed spacer and membrane surface coupon major elemental composition from the membrane autopsy.

Element	Value (mg/m^2^)
**Fe**	191.1
**Mg**	17.6
**Ni**	13.7
**Mn**	1.4

## Data Availability

Not applicable.

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
