# Peer review of "Seawater Reverse Osmosis Performance Decline Caused by Short-Term Elevated Feed Water Temperature"

_membranes, 2022, doi:10.3390/membranes12080792_

Round 1

Reviewer 1 Report

This paper presents how short term high and low temperature affect the performance of membrane. Some of the results are quite interesting. However, fouling effect is not considered in this study. Can the authors at least use mathematical equations to describe the impact of temperature on fouling studied?

Reviewer 2 Report

This manuscript investigated the temperature effect on RO membrane performance during short-term operation and emphasized the importance of developing new membranes for varied temperatures. However, it should be improved in some aspects and revisions are required before its publication in Membranes.

 Comments:

1)      In Table 1, please check the values of charge for each ion, especially sulfate and carbonate;

2)      The first paragraph of “3. Results” section should be reorganized and transferred to Introduction section.

3)      In figure 2C, why is there a sudden decrease in the pH of SWRO brine? Please explain.

4)      In figures 3 and 4, the definition and calculation of normalized flux should be given before the discussion on it.

5)      The same case in figure 5, there is no definition or calculation method on salt rejection in the manuscript.

6)      Please correct the typo in figure 6, NDP (not NSP) the normalized different pressure.

7)      How to explain the minor decrease of 2.5% in NPF (Line 271) and a reduction 0.2% in NSP (Line 276) if they are measurement errors? There are no error bars in all of the figures.

8)      The layer thickness of the virgin PA membrane (Line 294) should be measured in this study and compared to the used membrane because the data provided in the reference might be different from this study.

9)      In the conclusions, the operating cost of different plants are given. How did the authors get these data?

Reviewer 3 Report

In this manuscript, the authors presented an RO application case under seawater surroundings with a larger temperature difference. This work has high application value and meets the standard of this journal. However, a major revision is still needed before being accepted. The specific comments are as follows.

1. Please add experiments or cite related references to discuss the influence of water temperature change on membrane properties theoretically.

2. The abscissa axis have a confusing order in the figure 9 and the Average Specific Energy Consumption (SEC) before, during, and after short-term (7 day) have not clearly marked in the picture.

Round 2

Reviewer 1 Report

Accept

Reviewer 2 Report

OK.

Reviewer 3 Report

After revised, this manuscript meets the standard of this journal and can be accepted.